# Clinical Association between Gout and Parkinson’s Disease: A Nationwide Population-Based Cohort Study in Korea

**DOI:** 10.3390/medicina57121292

**Published:** 2021-11-24

**Authors:** Ji Hyoun Kim, In Ah Choi, Aryun Kim, Gilwon Kang

**Affiliations:** 1Division of Rheumatology, Department of Internal Medicine, Chungbuk National University Hospital, Cheongju 28644, Korea; jeehyounee@cbnuh.or.kr (J.H.K.); iachoi@cbnu.ac.kr (I.A.C.); 2Department of Internal Medicine, College of Medicine, Chungbuk National University, Cheongju 28644, Korea; 3Department of Neurology, Chungbuk National University Hospital, Cheongju 28644, Korea; mypioneer97@gmail.com; 4Department of Health Information and Management, College of Medicine, Chungbuk National University, Cheongju 28644, Korea

**Keywords:** gout, uric acid, Parkinson’s disease

## Abstract

*Background and Objectives*: This retrospective cohort study aimed to investigate the association between gout and Parkinson’s disease (PD) in Korea. *Materials and Methods*: Overall, 327,160 patients with gout and 327,160 age- and sex-matched controls were selected from the Korean National Health Insurance Service (NHIS) database. PD incidence was evaluated by reviewing NHIS records during the period from 2002 to 2019. Patients with a diagnosis of gout (International Classification of Diseases-10 (ICD-10), M10) who were prescribed medications for gout, including colchicine, allopurinol, febuxostat, and benzbromarone for at least 90 days were selected. Patients with PD who were assigned a diagnosis code (ICD-G20) and were registered in the rare incurable diseases (RID) system were extracted. *Results*: During follow-up, 912 patients with gout and 929 control participants developed PD. The incidence rate (IR) of overall PD (per 1000 person-years) was not significantly different between both groups (0.35 vs. 0.36 in gout and control groups, respectively). The incidence rate ratio (IRR) was 0.98 (95% CI: 0.89–1.07). The cumulative incidence of PD was not significantly different between the groups. No association between gout and PD was identified in univariate analysis (HR = 1.00, 95% CI: 0.91–1.10, *p* = 0.935). HR increased significantly with old age (HR = 92.08, 198, and 235.2 for 60–69 years, 70–79 years, and over 80 years, respectively), female sex (HR = 1.21, 95% CI: 1.07–1.37, *p* = 0.002), stroke (HR = 1.95, 95% CI: 1.76–2.16, *p* < 0.001), and hypertension (HR = 1.16, 95% CI: 1.01–1.34, *p* = 0.04). Dyslipidemia exhibited an inverse result for PD (HR = 0.6, 95% CI: 0.52–0.68, *p* < 0.001). *Conclusions*: This population-based study did not identify an association between gout and PD. Age, female sex, stroke, and hypertension were identified as independent risk factors for PD, and dyslipidemia demonstrated an inverse result for PD.

## 1. Introduction

Parkinson’s disease (PD) is a chronic neurodegenerative disease caused by prominent dopaminergic neuron loss and the accumulation of α-synuclein-containing Lewy bodies and Lewy neurites [1]. The characteristic clinical symptoms of PD are motor symptoms, including tremors, rigidity, and bradykinesia [2]. PD is the second most-common neurodegenerative disease, after Alzheimer’s disease [3], and affects 1–2% of individuals over 65 years of age [4], with a reported prevalence of 315 per 100,000 persons of all ages [4]. A previous study reported that the prevalence of PD was lower in Asian populations than in other ethnic groups [5]. Nevertheless, the prevalence of PD in Korea was reported to be 156.9 per 100,000 persons in 2012 and 181.3 per 100,000 persons in 2015, highlighting the increase in PD diagnosis over the past 3 years [6].

Gout is a chronic inflammatory disease characterized by severe pain caused by joint and soft tissue inflammation [7]. Previous studies have reported that uric acid may possess antioxidant properties and that low serum uric acid levels are associated with an increased risk of several neurodegenerative diseases [8,9,10]. Further, hyperuricemia has been reported to reduce the risk of PD [11] and inflammatory arthritis, including gout, has been identified as a risk factor for PD. However, the relationship between gout and PD is still unclear. This study aimed to investigate the association between gout and PD [12,13,14] by using large-scale data.

## 2. Materials and Methods

### 2.1. Database and Study Population

This retrospective cohort study used data from the Korean National Health Insurance Service (NHIS) database. The NHIS was launched in 2000 as the single insurer covering the entire population of Korea by integrating 375 insurance associations [15]. The NHIS comprises the eligibility database, national health screening database, basic demographic variables (i.e., age, sex, residential area, type of health insurance, etc.), International Classification of Diseases-10 (ICD-10) disease codes, health care utilization (diagnosis, length of stay, treatment costs, and services received), and prescription records (drug code, days prescribed, and daily dosage) [16].

### 2.2. Study Cohort Selection and Parkinson’s Disease Assessment

Data from the NHIS database covering the period from 2002 to 2019 were used. Data from 2002 were excluded as a wash-out period for new patient extraction, and data from 2018 and 2019 were also excluded to secure a follow-up period of at least 2 years. In total, 628,565 participants with a diagnosis of gout (ICD-10, M10) who were prescribed medications for gout, such as colchicine, allopurinol, febuxostat, and benzbromarone for at least 90 days were selected from the NHIS database. In this study, in order to improve the reliability of the diagnosis of patients with gout, the gout diagnosis ICD code was matched with the gout-specific medication code. As this was a study that used diagnostic codes and medication prescription codes, laboratory findings, including uric acid levels, could not be confirmed.

Patients with PD who were assigned a diagnosis code (ICD-G20) and were registered in the rare incurable diseases (RID) system were extracted. Patients with secondary Parkinsonism (G21, G22) or dyskinesia due to other causes (G23–G25) were excluded. The Korean government provides financial support to patients with rare, incurable rheumatic diseases that are specially registered and have been managed since 2009. To be eligible for registration in the RID system, patients must meet the diagnostic criteria of each RID and are carefully reviewed by the corresponding healthcare institution. Only a neurologist or neurosurgeon can register patients with PD according to the United Kingdom PD Society Brain Bank criteria on the RID system. The diagnosis is also made by excluding other forms of secondary parkinsonism. Therefore, diagnoses of patients with PD registered in the RID system are considered highly reliable.

In this study, 215,711 patients with cancer, 85,331 patients with end-stage renal disease (ESRD), and 363 previous patients with PD were removed. A final total of 327,160 new gout patients were extracted. The control cohort was defined as patients who had never received a gout diagnostic code and had never been administered any gout medication. Age and sex (1:1) propensity score matching was performed in the selected control cohort. The flow chart for study cohort selection is presented in Figure 1.

### 2.3. Statistical Analysis

The baseline demographics and clinical characteristics of the patients with gout and the controls at baseline were analyzed using a chi-square test or a Student’s *t*-test. The Cox proportional hazards model was used to evaluate the effect of gout on the risk of PD, and the results are presented as hazard ratios (HRs) with 95% confidence intervals (CIs) using different adjustment models for potential confounders. There was no multicollinearity between variables, and the proportional hazard assumption was met. Cumulative PD incidence was determined using the Kaplan–Meier method and compared between the two cohorts using the log-rank test. Analyses were performed using the SAS statistical package version 9.4 (SAS Institute Inc., Cary, NC, USA) and R version 3.4.3 (R Foundation for Statistical Computing, Vienna, Austria). A *p*-value < 0.01 was considered statistically significant.

## 3. Results

The database included medical information of almost all Korean individuals enrolled in the NHIS database. In total, 630,000 new patients diagnosed with gout were identified during the 18-year period from 2002 to 2019. A final total of 327,160 patients with gout satisfied the study criteria. Our analysis revealed the distribution of patients with gout in Korea according to age, with the following approximate numbers: <30 years (n = 30,000), 30–39 years (n = 75,000), 40–49 years (n = 85,000), 50–59 years (n = 70,000), 60–69 years (n = 40,000), 70–79 years (n = 20,000), and 80–89 years (n = 6000). The age group with the highest prevalence was the 40–49 years age group. Most patients with gout were male (304,162; 93%). No significant differences were observed in the mean age and sex distribution between the patients with gout and the controls, indicating that the groups were well matched. Participants with diabetes comprised 40% and 57% of the control and gout groups, respectively *(p <* 0.001). Hypertension (40% in the control group vs. 65% in the gout group), dyslipidemia (62% in the control group vs. 87% in the gout group), ischemic heart disease (22% in the control group vs. 31% in the gout group), and stroke (8.5% in the control group vs. 10.7% in the gout group) were significantly more common in patients with gout than in controls. The demographics and clinical characteristics of the study cohort at baseline are presented in Table 1.

Table 2 presents the incidence of PD in patients with gout and in the control group. Overall, 912 patients with gout and 929 control participants developed PD. The incidence rate (IR) of overall PD (per 1000 person-years) was not significantly different between the two groups (0.35 in the gout group vs. 0.36 in the control group). The incidence rate ratio (IRR) was 0.98 (95% CI: 0.89–1.07). A stratified analysis by sex and age did not reveal any significant differences between the groups (Table 2). In the analysis of the cumulative incidence of PD, no significant between-group differences were observed (Figure 2).

The association between gout and PD was not significant (HR = 1.00, 95% CI: 0.91–1.10, *p* = 0.935) (Table 3). Therefore, other potential risk factors for PD were investigated. The most significant result was the age of patients. We identified a significant association between the incidence of PD and aging (HR = 92.08, 198, and 135.3 in 60–69 years, 70–79 years, and >80 years, respectively). Other factors that affected the incidence of PD were female sex (HR = 1.21, 95% CI: 1.07–1.37, *p* = 0.0024), stroke (HR = 1.95, 95% CI: 1.76–2.16, *p* < 0.001), and hypertension (HR = 1.16, 95% CI: 1.01–1.34, *p* = 0.040). Notably, dyslipidemia exhibited inverse results for PD (HR = 0.6, 95% CI: 0.52–0.68, *p* < 0.001) (Table 3).

We divided gout patients into male and female groups and performed a sub-analysis on comorbidities affecting PD. The risk was highest in stroke, HR 2.15 (95% CI 1.91–2.41, *p* < 0.0001) in the male group and HR 1.39 (95% CI 1.12–1.73, *p* = 0.0025) in the female group. In the female group, the risk of hypertension increased to HR 1.46 (95% CI 0.96–2.22, *p* = 0.0793), but there was no significance (Appendix A).

## 4. Discussion

The purpose of this study was to determine the association between gout and the incidence of PD. We did not observe any significant differences in the incidence and risk of PD between the patients with gout and the controls using large-scale data. Previous studies have reported an association between hyperuricemia and PD. An analysis of US Medicare data of 1.7 million individuals from 2006 to 2012 revealed that hyperuricemia increased the risk of incidental PD (HR = 1.27, 95% CI: 1.16–1.39) in individuals between 65 and 75 years of age [17]. In contrast, a retrospective study, using Taiwan’s National Health Insurance Research Database, that analyzed 7900 patients with gout and 1:1-matched control participants without gout did not reveal any significant results (HR = 1.01, 95% CI: 0.93–1.31) [18]. Similar to the Taiwanese study, we did not identify a significant association between gout and PD. However, PD is a relatively rare disease, with a lower prevalence in Asian populations than in other ethnic groups [5]. Therefore, these discrepant results may be due to genetic differences between Asians and other ethnic groups. Additionally, since our study only extracted patients with PD registered in the RID system, the patients included in this study predominantly comprised those who visited tertiary hospitals. Therefore, the incidence of PD may have been underestimated. In addition, in order to increase the reliability of the diagnosis for gout and to differentiate it from other studies, we defined gout groups as those who had a diagnosis code and continued gout medication for 3 months or more. Therefore, it can be expected that the patients with a relatively well-controlled uric acid level were also included in this study. It was inappropriate to evaluate the protective effect of uric acid on the development of PD. Further studies are needed to address this issue.

In our study, the incidence of cardiovascular comorbidities, including hypertension, dyslipidemia, ischemic heart disease, and stroke, were significantly increased among patients with gout compared to the control group. These results were consistent with previous studies [19,20,21,22].

In this study, we did not identify an association between gout and PD after adjusting for age, sex, and comorbidities. PD is a complex neurodegenerative disease and it can be affected by many other factors other than gout. Therefore, we investigated other potential factors affecting PD. The Cox proportional hazards model revealed that old age, female sex, stroke, and hypertension were independent risk factors for PD.

Han et al. [6] reported that the prevalence and incidence of PD were higher in women than in men. Other studies on Asian populations in Japan, China, and Taiwan revealed consistent results, with a reported female preponderance or no sex-related differences in the occurrence of PD [23,24,25]. In our study, the result identified female sex as a risk factor for PD, although only 7% of the study’s cohort was female. Conversely, studies in Europe and North/South America have reported opposing results, suggesting that sex differences may exist according to race/ethnicity [26,27,28,29].

We conducted a sub-analysis by sex group for risk factors affecting PD using the Cox proportional hazards model. The analysis result revealed that stroke was still an important risk factor, and that hypertension also tended to increase risk for PD in the female group. Therefore, we suggest that controlling hypertension, a modifiable risk factor, and preventing stroke can help lower the risk of PD, especially in females, who comprise the risk group.

Our study results revealed an inverse relationship between PD and dyslipidemia. The association between dyslipidemia and PD is controversial [30,31,32,33,34]. In this study, the HR of dyslipidemia for PD risk was 0.6 (95% CI 0.52–0.68, *p* < 0.001) in the whole of the patients, HR 0.6 in the male group (95% CI 0.52–0.69, *p* < 0.0001) and HR 0.59 in the female group (95% CI 0.42–0.82, *p* = 0.0017). These results were consistent with two recent large cohort studies [32,33]. A possible explanation is as follows: cholesterol is involved in critical biological functions, including cellular repair, degeneration, and acts as a neurosteroid precursor. Potential beneficial roles of higher cholesterol levels in other neurodegenerative disorders have also been reported [35,36,37,38,39]. Nevertheless, the mechanisms of these effects have yet to be elucidated. Another potential mechanism involves the anti-inflammatory actions of statins, which reduce oxidative stress and neuro-inflammation [40,41].

A previous study reported several risk factors and protective factors for PD. For instance, cigarette smoking, coffee drinking, vitamin E intake, β2-adrenoceptor agonist, and gout were suggested to exert protective effects [42]. Indeed, several reports have indicated that uric acid is a possible protective factor for PD [43,44]. However, the mechanisms underlying the inverse association between serum uric acid and PD remain unclear.

Uric acid is a product of purine metabolism and accounts for approximately 60% of free radical scavenging activity in human serum. Based on results from in vitro and in vivo experimental studies, uric acid may act as a neuroprotective agent by scavenging peroxynitrite-derived radicals, which are involved in the pathogenesis of neurodegenerative diseases [43,44]. Uric acid has also been reported to have iron-chelating properties. Iron is an essential element in cell metabolism and is involved in myelination and neurotransmission. The accumulation of iron in the substantia nigra, a region critically involved in PD, plays an important role in the death of dopaminergic neurons [45,46]. However, we did not examine this hypothesis in the current study.

Our study has several limitations. First, we were unable to collect data regarding participants’ serum uric acid levels. Second, we did not investigate other environmental factors that could affect PD, such as residence, income, education level, drug use, smoking or drinking history, obesity, and other comorbidities. Furthermore, we included only patients who were administered gout-specific medications to increase the reliability of the diagnosis of gout. This is a limitation, and further studies on the effects of gout medications are needed. Nevertheless, this study represents the largest analysis of the current dataset so far. Further, the study results revealed the cumulative incidence of gout according to major age groups, sex, and underlying comorbidities because patients who recently developed gout were enrolled. Furthermore, to increase the reliability of a diagnosis of PD, we extracted patients with incidental PD enrolled in the RID system. Additionally, by performing a subgroup analysis, we were able to elucidate other risk factors and protective factors for PD.

## 5. Conclusions

This population-based study did not identify an association between gout and PD. Clinical characteristics, including frequency according to age group and sex distribution in domestic patients with gout, were confirmed. Further, our analysis identified the percentage of cardiovascular and metabolic comorbidities. Age, female sex, stroke, and hypertension were identified as independent risk factors for PD, and dyslipidemia exhibited an inverse relationship with PD. Further research is warranted to verify our findings.

## Figures and Tables

**Figure 1 medicina-57-01292-f001:**
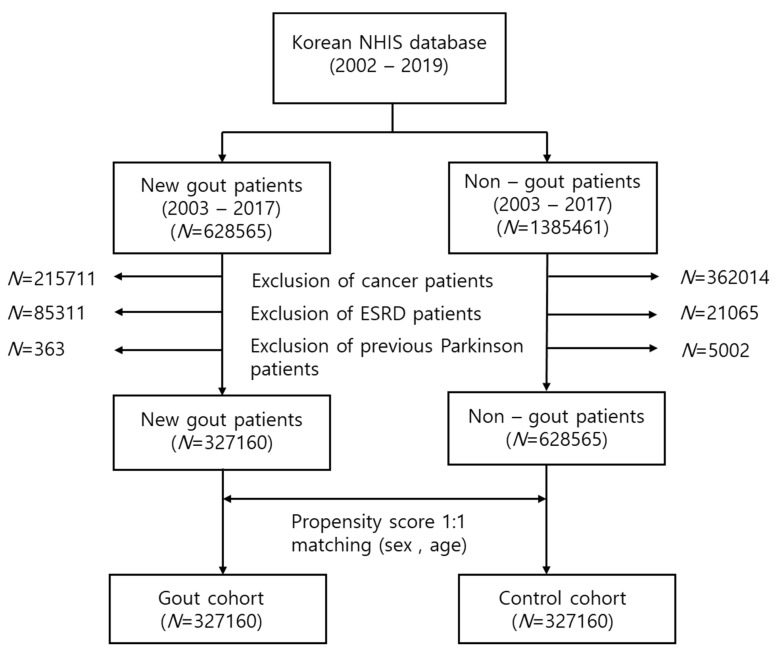
Flow chart for the selection of the study population from the National Health Insurance Service database. In total, 327,160 patients with gout and 327,160 control participants were compared via propensity score matching.

**Figure 2 medicina-57-01292-f002:**
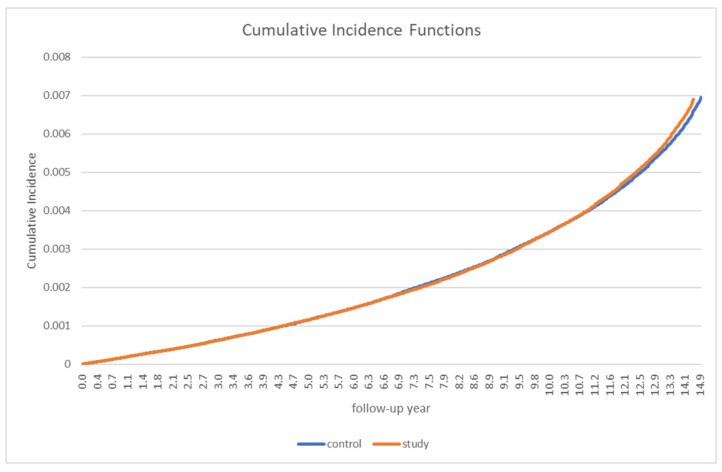
Cumulative Parkinson’s disease incidence in patients with gout and the controls.

**Table 1 medicina-57-01292-t001:** Demographics and clinical characteristics of the study cohort.

Variables	Gout Patients	Controls	*p*-Value
	(*N* = 327,160)	(*N* = 327,160)	
Sex Male Female	304,162 22,998	304,162 22,998	
Age (years)			
<30	30,581	30,581	
30–39	75,265	75,265	
40–49	85,730	85,730	
50–59	70,912	70,912	
60–69	39,063	39,063	
70–79	19,638	19,638	
≥80	5.971	5.971	
Underlying diseases			
Hypertension	211,097 (64.52%)	140,890 (43.06%)	<0.001
Diabetes	186,198 (56.91%)	130,351(39.84%)	<0.001
Dyslipidemia	284,794 (87.05%)	203,823 (62.3%)	<0.001
Ischemic heart disease	102,591(31.36%)	73,162 (22.36%)	<0.001
Stroke	34,920 (10.67%)	27,777 (8.49%)	<0.001

Values are presented as number of patients (%).

**Table 2 medicina-57-01292-t002:** Incidence of Parkinson’s disease in patients with gout and in the controls, stratified by age group and sex.

Variables	Gout (*N* = 327,160)			Controls (*N* = 327,160)			IRR (95% CI)
	*N*, %	PY	IR	*N*, %	PY	IR	
Overall	912	6544.07	0.35	929	6676.18	0.36	0.98 (0.89–1.07)
Sex							
Male	718	5437.98	0.29	748	5541.36	0.31	0.98 (0.88–1.08)
Female	194	1106.09	1.32	181	1134.82	1.19	1.09 (0.9–1.35)
Age, years							
20–29	2	8.18	0.01	3	11.45	0.01	0.67 (0.11–4.0)
30–39	14	87.16	0.02	17	139.21	0.03	0.82 (0.41–1.67)
40–49	60	441.84	0.08	66	524.01	0.09	0.91 (0.64–1.28)
50–59	164	1363.84	0.28	170	1334.36	0.3	0.95 (0.77–1.18)
60–69	325	2550.98	1.10	312	2475.13	1.07	1.03 (0.88–1.2)
70–79	277	1779.07	0.56	299	1952.53	2.41	0.91 (0.78–1.08)
≥80	70	313.01	2.53	62	239.48	2.40	1.05 (0.75–1.48)

PY: 1000 person-years; IR: incidence ratio; IRR: incidence rate ratio; CI: confidence interval.

**Table 3 medicina-57-01292-t003:** Cox proportional hazards model of risk factors for Parkinson’s disease.

	Unadjusted HR (95% CI)	*p*-Value	Adjusted HR (95% CI)	*p*-Value
Gout	0.98 (0.89–1.07)	0.602	1.00 (0.91–1.10)	0.935
Female sex	4.731 (4.22–5.30)	<0.001	1.21 (1.07–1.37)	0.002
Age				
30–39	2.29 (0.89–5.9)	0.085	2.33 (0.91–6.00)	0.079
40–49	7.58 (3.10–18.54)	<0.001	7.7 (3.15–18.83)	<0.001
50–59	27.63 (43.35–251.96)	<0.001	26.95 (11.12–65.34)	<0.001
60–69	104.51 (43.35–251.96)	<0.001	92.08 (38.03–222.93)	<0.001
70–79	256.07 (106.17–617.61)	<0.001	197.5 (81.39–479.21)	<0.001
≥80	343.41 (140.52–839.22)	<0.001	235.26 (95.53–579.38)	<0.001
Hypertension	3.59 (3.17–4.03)	<0.001	1.16 (1.01–1.34)	0.040
Diabetes	2.02 (1.83–2.22)	<0.001	0.93 (0.83–1.04)	0.216
Dyslipidemia	1.25 (1.11–1.40)	<0.001	0.6 (0.52–0.68)	<0.001
Ischemic heart disease	2.37 (2.17–2.60)	<0.001	0.99 (0.90–1.1)	0.998
Stroke	5.71 (5.2–6.27)	<0.001	1.95 (1.76–2.16)	<0.001

HR: hazard ratio (using the Cox proportional hazards model), CI: confidence interval.

## Data Availability

Data cannot be shared publicly because they belong to the National Health Insurance Service (NHIS). There are ethical restrictions on sharing such a data set because these data contain potentially identifying or sensitive patient information. To request data from NHIS, researchers have to apply during the recruitment period and submit a research proposal. Raw data was available to researchers upon reasonable academic request and with the permission of the Korean NHIS Institutional Data Access (https://nhiss.nhis.or.kr/bd/af/bdafa001lv.do). The authors had no special access privileges.

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
