# Peer review of "Clinical Association between Gout and Parkinson’s Disease: A Nationwide Population-Based Cohort Study in Korea"

_medicina, 2021, doi:10.3390/medicina57121292_

Round 1
Reviewer 1 Report
The article is well-written, grammatic is ok, good methods, and discussion.
However, I have some comments:
1) The article is missing something important about ‘‘what does this manuscript adds new for the literature?’’
2) The article cited by Hu et al. 2020 already described these negative results in a Taiwanese population. And it is very similar in means of methods. What do we have different in this proposed manuscript? Hu LY, Yang AC, Lee SC, You ZH, Tsai SJ, Hu CK, Shen CC. Risk of Parkinson’s disease following gout: a population-based retrospective cohort study in Taiwan. BMC neurology. 2020 Dec;20(1):1-7.
3) PD diagnosis was done by at least two board-certified neurologists (diagnostic criteria)? And about gout how was the diagnosis done?
4) Drugs and other medications being used were analyzed?
5) A population-based cohort study is still an observational study and many other aspects should be listed as drawbacks such as records that were not designed for the study, absence of data on potential confounding factors, and others.
6) In the title should be written the country from which population the data was extracted
7) How much time these patients were followed?
8) What were the confounder variables? Please, specify the adjustment models. ‘‘using different adjustment models for potential confounders’’.
9) If we analyze the sample size, the p-value chosen is large. Why did you not choose a lower p-value? https://www.researchgate.net/post/Should-my-alpha-be-set-to-05-or-01
IDEA: Even though the authors describe that the data is not available, they could anonymize and provide it as supplemental material. It would greatly improve the quality of the manuscript. And possibly re-analyzations of the data. ‘’To remain relevant to research needs, observational cohort studies must also allow access to their rich resources.’’ Sorlie P, Wei GS. Population-based cohort studies: still relevant? Journal of the American College of Cardiology. 2011 Nov 1;58(19):2010-3.
Author Response
Comments and Suggestions for Authors (Reviewer #1)
The article is well-written, grammatic is ok, good methods, and discussion.
However, I have some comments:
1)The article is missing something important about ‘‘what does this manuscript adds new for the literature?’’
-> Response to reviewer #1
Thanks for your comment.
In our study, we analyzed large-scale data in order to clarify the debate regarding whether there was an association between gout and PD. We believe our study provides strong evidence that there is no direct association between gout and PD. We have revised the introduction and discussion sections to clarify the purpose and strength of our research. The changes made are as follows:
=> [Lines 52-54] However, the relationship between gout and PD is still unclear. This study aimed to investigate the association between gout and PD [12-14] by using large-scale data.
=> [Lines 158-159] We did not observe any significant difference in the incidence and risk of PD between patients with gout and controls using large-scale data.
2) The article cited by Hu et al. 2020 already described these negative results in a Taiwanese population. And it is very similar in means of methods. What do we have different in this proposed manuscript? Hu LY, Yang AC, Lee SC, You ZH, Tsai SJ, Hu CK, Shen CC. Risk of Parkinson’s disease following gout: a population-based retrospective cohort study in Taiwan. BMC neurology. 2020 Dec;20(1):1-7.
-> Response to reviewer #1
Thanks for your very important comments.
The Taiwanese study was conducted from 2000 to 2013, while our study is supported by long-term data from 2002 to 2019 (including a 1-year wash-out period). In the Taiwanese study, the number of patients with gout was 7900, and in our study, we enrolled 327,160 patients with gout; hence, the number of patients in this study was 40 times larger than that of the Taiwanese study. Therefore, although the Taiwanese study is novel, our research was more powerful.
Further, we tried to increase the diagnostic value of gout using the ICD-code with prescriptions of gout specific medications. For the diagnosis of PD, only patients registered in the RID system were selected. Patients enrolled in the RID system in Korea must meet the diagnostic criteria of each RID and are carefully reviewed by the corresponding healthcare institution’s specialists. Therefore, the high reliability of the diagnoses is a strength of our study.
3) PD diagnosis was done by at least two board-certified neurologists (diagnostic criteria)? And about gout how was the diagnosis done?
-> Response to reviewer #1
As described above, the RID system is practiced in Korea. Only a neurologist or neurosurgeon can register a diagnosis of PD on the RID system for patients that meet the United Kingdom PD Society Brain Bank criteria. The diagnosis is also made by excluding other secondary forms of parkinsonism. Patients with PD are carefully managed through the RID system; hence, this system is very reliable.
However, using only ICD-codes for a diagnosis of gout is inaccurate in Korea. We matched ICD codes with gout-specific medication codes to improve the accuracy of the diagnosis of gout. We added this information to in the methods section as follow:
Although this has the advantage of increasing the reliability of the diagnosis of gout, additional research is needed because only patients with gout on medications were included in this study; therefore, whether medications affected the results could not be ascertained. The limitations related to this are as follows and mentioned in discussion.
=> [Lines 72-74] Data from the NHIS database covering the period from 2002 to 2019 were used. Data from 2002 were excluded as a wash-out period for new patient extraction, and data from 2018 and 2019 were also excluded to secure a follow-up period of at least 2 years. In total, 628,565 participants with a diagnosis of gout (ICD-10, M10) who were prescribed medications for gout, such as colchicine, allopurinol, febuxostat, and benzbromarone for at least 90 days were selected from the NHIS database. In this study, in order to improve the reliability of the diagnosis of patients with gout, the gout diagnosis ICD code was matched with the gout-specific medication code.
=> [Lines 83-84] Patients with PD who were assigned a diagnosis code (ICD-G20) and were registered in the rare incurable diseases (RID) system were extracted. Patients with secondary Parkinsonism (G21, G22) or dyskinesia due to other causes (G23-G25) were excluded. The Korean government provides financial support to patients with rare, incurable rheumatic diseases that are specially registered and have been managed since 2009. To be eligible for registration in the RID system, patients must meet the diagnostic criteria of each RID and are carefully reviewed by the corresponding healthcare institution. Only a neurologist or neurosurgeon can register patients with PD according to the United Kingdom PD Society Brain Bank criteria on the RID system. The diagnosis is also made by excluding other forms of secondary parkinsonism. Therefore, the diagnosis of patients with PD registered in the RID system is considered highly reliable.
=> [Lines 212-217] Our study has several limitations. First, we were unable to collect data regarding participants’ serum uric acid levels. Second, we did not investigate other environmental factors that could affect PD, such as residence, income, education level, drug use, smoking, drinking history, obesity, and other comorbidities. Furthermore, we included only patients who were administered gout-specific medications to increase the reliability of the diagnosis of gout. This is a limitation, and further studies on the effects of gout medications are needed.
4) Drugs and other medications being used were analyzed?
-> Response to reviewer #1
Drugs or other medications that could affect the development of PD were not analyzed. This part is also mentioned as a limitation in the discussion.
=> [Lines 212-214] Our study has several limitations. First, we were unable to collect data regarding participants’ serum uric acid levels. Second, we did not investigate other environmental factors that could affect PD, such as residence, income, education level, drug use, smoking, drinking history, obesity, and other comorbidities.
5) A population-based cohort study is still an observational study and many other aspects should be listed as drawbacks such as records that were not designed for the study, absence of data on potential confounding factors, and others.
-> Response to reviewer #1
Thanks for your comment. Due to data limitations, such as residence, income, education level, drug use, smoking, drinking history, obesity, and other comorbidities, the part that was related to potential confounding factors could not be analyzed, and this is a limitation of this study. We have added this to the discussion section.
=> [Lines 212-214] Our study has several limitations. First, we were unable to collect data regarding participants’ serum uric acid levels. Second, we did not investigate other environmental factors that could affect PD, such as residence, income, education level, drug use, smoking, drinking history, obesity, and other comorbidities.
6) In the title should be written the country from which population the data was extracted
Response to reviewer #1
I agree with your opinion. We have changed the title to “Clinical association between gout and Parkinson’s disease: A nationwide population-based cohort study in Korea”.
7) How much time these patients were followed?
Response to reviewer #1
Our data was obtained from an open patient cohort in which new patients are added yearly, and the follow-up period varies from patient to patient. The shortest follow-up period was 2 years (2017–2019), while the longest was 17 years (2003–2019).
8) What were the confounder variables? Please, specify the adjustment models. ‘‘using different adjustment models for potential confounders’’.
Response to reviewer #1
We apologize for the misleading statement. This seems to be a misunderstanding caused by the fact that univariate analysis was included in the title of Table 3. The adjusted HR in Table 3 was obtained using the cox proportional hazard model and were adjusted for all other listed variables. The inappropriate table title has been corrected. The changes are as follows.
=> Table 3. Cox proportional hazard model of the risk factors for Parkinson’s disease
9) If we analyze the sample size, the p-value chosen is large. Why did you not choose a lower p-value? https://www.researchgate.net/post/Should-my-alpha-be-set-to-05-or-01
Response to reviewer #1
Thanks for your helpful comments. According to your opinion, the significance criterion was changed to p<0.01. However, even after the significance criterion was lowered, the conclusion did not change.
=> [Lines 109] A p-value of < 0.01 was considered statistically significant.
IDEA: Even though the authors describe that the data is not available, they could anonymize and provide it as supplemental material. It would greatly improve the quality of the manuscript. And possibly re-analyzations of the data. ‘’To remain relevant to research needs, observational cohort studies must also allow access to their rich resources.’’ Sorlie P, Wei GS. Population-based cohort studies: still relevant? Journal of the American College of Cardiology. 2011 Nov 1;58(19):2010-3.
Response to reviewer #1
Health insurance data is prohibited from sharing raw data. If the requested data is raw data, it cannot be shared due to the regulations of the data provider. To request data from NHIS, researchers have to apply during the recruitment period and need to also submit a research proposal. The committee reviews the proposals and then selects a few researchers to use and analyze the data. Data access applications for the national health insurance data are available on the NHIS data sharing website (https://nhiss.nhis.or.kr/bd/ou/bdoua001lv.do). Raw data are available to researchers upon reasonable academic request and with the permission of the Korean NHIS Institutional Data Access (http://nhiss.nhis.or.kr). The authors had no special access privileges.
Reviewer 2 Report
The manuscript of Hyoun Kim et al is a retrospective observational study on the association of gout and Parkinson’s disease (PD) in 654,320 adults from Korea. The topic is of interest and the study is well written. However, I have some comments:
In the present study hypertension directly correlates with PD. Hyperuricemia has been shown to strongly associate with hypertension in several studies. I would further investigate the relationship between gout and hypertension.
Sanchez-Lozada LG, Rodriguez-Iturbe B, Kelley EE, Nakagawa T, Madero M, Feig DI, et al. Uric Acid and Hypertension: An Update With Recommendations. Am J Hypertens. 2020 Jul 18;33(7):583-594. doi: 10.1093/ajh/hpaa044. Erratum in: Am J Hypertens. 2020 Dec 31;33(12):1150. PMID: 32179896; PMCID: PMC7368167.
Kutzing MK, Firestein BL. Altered uric acid levels and disease states. J Pharmacol Exp Ther. 2008 Jan;324(1):1-7. doi: 10.1124/jpet.107.129031. Epub 2007 Sep 21. PMID: 17890445.
Piani F, Cicero AFG, Borghi C. Uric Acid and Hypertension: Prognostic Role and Guide for Treatment. J Clin Med. 2021 Jan 24;10(3):448. doi: 10.3390/jcm10030448. PMID: 33498870; PMCID: PMC7865830.
- All participants were under treatment for gout, I think this may have biased the results. I would perform sensitivity analysis for treatment groups (e.g. patients treated with allopurinol and risk of PD, and so on). Since some hypouricemic compounds have shown antihypertensive properties, it would be interesting to know if participants with gout treated with such compounds have a lower risk for development of PD. This may explain the absence of a significant association between gout and PD.
- The Authors correctly addressed the implications of uric acid in neurodegenerative diseases. However, they were not able to collect data on participants’ serum uric acid levels. I think this is a major limitation of the study and should be disclosed in the Methods section.
Author Response
Comments and Suggestions for Authors (Reviewer #2)
The manuscript of Hyoun Kim et al is a retrospective observational study on the association of gout and Parkinson’s disease (PD) in 654,320 adults from Korea. The topic is of interest and the study is well written. However, I have some comments:
1) In the present study hypertension directly correlates with PD. Hyperuricemia has been shown to strongly associate with hypertension in several studies. I would further investigate the relationship between gout and hypertension.
Sanchez-Lozada LG, Rodriguez-Iturbe B, Kelley EE, Nakagawa T, Madero M, Feig DI, et al. Uric Acid and Hypertension: An Update With Recommendations. Am J Hypertens. 2020 Jul 18;33(7):583-594. doi: 10.1093/ajh/hpaa044. Erratum in: Am J Hypertens. 2020 Dec 31;33(12):1150. PMID: 32179896; PMCID: PMC7368167.
Kutzing MK, Firestein BL. Altered uric acid levels and disease states. J Pharmacol Exp Ther. 2008 Jan;324(1):1-7. doi: 10.1124/jpet.107.129031. Epub 2007 Sep 21. PMID: 17890445.
Piani F, Cicero AFG, Borghi C. Uric Acid and Hypertension: Prognostic Role and Guide for Treatment. J Clin Med. 2021 Jan 24;10(3):448. doi: 10.3390/jcm10030448. PMID: 33498870; PMCID: PMC7865830.
Response to reviewer #2
Thanks for your helpful comments. In this study, we confirmed that hypertension was an independent risk factor for PD after adjusting for all other listed variables using a cox proportional hazard model.
However, the incidence of cardiovascular comorbidities including hypertension, dyslipidemia, ischemic heart disease, and stroke significantly increased in the patient group with gout compared to the control group without gout. These results were consistent with previous studies as you mentioned. Therefore, we cited the above study as a reference and revised it as follows.
=> [ Lines 174-177] In our study the incidence of cardiovascular comorbidities including hypertension, dyslipidemia, ischemic heart disease, and stroke were significantly increased among patient with gout compared to the control group. These results were consistent with previous studies [19-22].
2)All participants were under treatment for gout, I think this may have biased the results. I would perform sensitivity analysis for treatment groups (e.g. patients treated with allopurinol and risk of PD, and so on). Since some hypouricemic compounds have shown antihypertensive properties, it would be interesting to know if participants with gout treated with such compounds have a lower risk for development of PD. This may explain the absence of a significant association between gout and PD.
Response to reviewer #2
Thank you for your important comments. This was one of the major limitations of this study. In Korea, using only the diagnostic code for gout is not sufficient to make a diagnosis of gout. Hence, patients with gout were extracted by combining the gout ICD code with the gout specific medication code. Therefore, analysis on the use of each drug for gout was not possible. Analysis of drugs with hypouricemic effect will be considered through follow-up studies in the future. These statements were added to the limitations mentioned in discussion section.
=> [Lines 214-217] Furthermore, we included only patients who were administered gout-specific medications to increase the reliability of the diagnosis of gout. This is a limitation, and further studies on the effects of gout medications are needed.
3)The Authors correctly addressed the implications of uric acid in neurodegenerative diseases. However, they were not able to collect data on participants’ serum uric acid levels. I think this is a major limitation of the study and should be disclosed in the Methods section.
Response to reviewer #2
Thank you for your comments. We have mentioned in the discussion section that one of the limitations of the data analyzed in this study was that uric acid levels could not be measured. In addition, we have described this in more details in the method section in line with your comment.
=>[Lines 74-76] As it was a study that used diagnostic codes and medication prescription codes, laboratory findings, including uric acid levels, could not be confirmed.
Round 2
Reviewer 1 Report
I appreciate the authors’ responses, but I am still not convinced that the manuscript brings anything new to the literature. The idea of PD and Gout started in 1805 with James Parkinson,1 after that we have a lot of articles trying to correlate both pathologies.2 Also, even though the manuscript by the author has a larger population and is long-term data did not change the same idea already present in the literature by Hu et al.3 In the reviewer's opinion, authors should overcome this problem by trying to analyze something different (variable) and include some new hypothesis or explaining why there is no association. This would greatly impact the quality of the manuscript.
It is worthy of mentioning that only 23.80% (10/42) references are from the last five years of science. It would be advised to be 60-70%.
1- https://en.wikipedia.org/wiki/James_Parkinson
2- https://pubmed.ncbi.nlm.nih.gov/?term=%28parkinson%29+AND+%28gout%29&sort=date&ac=no
3- Hu LY, Yang AC, Lee SC, You ZH, Tsai SJ, Hu CK, Shen CC. Risk of Parkinson's disease following gout: a population-based retrospective cohort study in Taiwan. BMC Neurol. 2020 Sep 8;20(1):338. DOI: 10.1186/s12883-020-01916-9. PMID: 32900384; PMCID: PMC7487828.
Author Response
- Response to Reviewer
Thank you very much for your valuable comment. We have done our best to respond to your comments during the given period. Recent studies have been also added as references.
- Another cause explaining why there is no association
In this study, in order to increase the reliability of the diagnosis for gout and to differentiate it from other studies, we defined gout group who had a diagnosis code and continued gout medication for 3 months or more. Therefore, it can be expected that the patients with relatively well controlled uric acid level also included in this study. In other words, it is possible that this factor affected not to lower the risk of PD.
- Additional subgroup analysis
PD is a complex neurodegenerative disease and it can be affected by many other factors other than gout. The association of PD with age, sex, hypertension, stroke, and dyslipidemia was significant in this study. We conducted sub-analysis by sex group for risk factors affecting PD using cox-proportional hazard model. The analysis result revealed that stroke was still important risk factor and hypertension also showed tendency to increased risk for PD in female group (Supplement table). The subgroup analysis results are as follows.
Supplement table. Cox proportional hazard model of risk factors for Parkinson’s disease by sex
|
|
Male group Adjusted HR (95% CI) |
P-value |
Female group, Adjusted HR (95% CI) |
P-value |
|
Gout |
0.96 (0.86-1.07) |
0.4485 |
1.19 (0.96-1.47) |
0.1176 |
|
Hypertension |
1.12 (0.96-1.3) |
0.1432 |
1.46 (0.96-2.22) |
0.0793 |
|
Diabetes |
0.95 (0.84-1.08) |
0.4145 |
0.84 (0.65-1.1) |
0.2036 |
|
Dyslipidemia |
0.6 (0.52-0.69) |
<.0001 |
0.59 (0.42- 0.82) |
0.0017 |
|
Ischemic heart disease |
0.98 (0.87-1.09) |
0.6557 |
1.07 (0.85- 1.34) |
0.5597 |
|
Stroke |
2.15 (1.91 -2.41) |
<.0001 |
1.39 (1.12-1.73) |
0.0025 |
We suggest that controlling hypertension, a modifiable risk factor, and preventing stroke can be helpful to lower the risk of PD, especially in the females, who are risk group.
- Latest researches regarding risk factor identified in this study
In addition, this study suggested that dyslipidemia reduced the risk of PD. The association between dyslipidemia and PD is controversial [30-34]. Our study result revealed that dyslipidemia reduced risk of PD, which is consistent with results from recent two large cohort studies [32,33].
A possible explanation is below. Cholesterol is involved in critical biological functions, including cellular repair, degeneration, and acts as a neurosteroid precursor. Potential beneficial roles of higher cholesterol levels in other neurodegenerative disorders have also been reported [35-39]. Nevertheless, the mechanisms of these effects are yet to be elucidated.
These contents have been added to the results (Supplement table) and discussion part.
=> [Results] We divided into subgroup by sex and performed sub-analysis on comorbidities affecting PD. The risk was highest in stroke, HR 2.15 (95% CI 1.91 -2.41, p<.0001) in the male group and HR 1.39 (95% CI 1.12-1.73, p= 0.0025) in the female group. In the female group, the risk of hypertension increased to HR 1.46 (95% CI 0.96-2.22, p=0.0793), but there was no significance (Supplement table)
Supplement table. Cox proportional hazard model of risk factors for Parkinson’s disease by sex
|
|
Male group Adjusted HR (95% CI) |
P-value |
Female group, Adjusted HR (95% CI) |
P-value |
|
Gout |
0.96 (0.86-1.07) |
0.4485 |
1.19 (0.96-1.47) |
0.1176 |
|
Hypertension |
1.12 (0.96-1.3) |
0.1432 |
1.46 (0.96-2.22) |
0.0793 |
|
Diabetes |
0.95 (0.84-1.08) |
0.4145 |
0.84 (0.65-1.1) |
0.2036 |
|
Dyslipidemia |
0.6 (0.52-0.69) |
<.0001 |
0.59 (0.42- 0.82) |
0.0017 |
|
Ischemic heart disease |
0.98 (0.87-1.09) |
0.6557 |
1.07 (0.85- 1.34) |
0.5597 |
|
Stroke |
2.15 (1.91 -2.41) |
<.0001 |
1.39 (1.12-1.73) |
0.0025 |
HR: hazard ratio (using the Cox proportional hazard model), CI: confidence interval
=>[Discussion] Therefore, these discrepant results may be due to genetic differences between Asians and other ethnic groups. Additionally, since our study only extracted patients with PD registered in the RID system, the patients included in this study predominantly comprised those who visited tertiary hospitals. Therefore, the incidence of PD may have been underestimated. In addition, in order to increase the reliability of the diagnosis for gout and to differentiate it from other studies, we defined gout group who had a diagnosis code and continued gout medication for 3 months or more. Therefore, it can be expected that the patients with relatively well controlled uric acid level also included in this study. In other words, it is possible that this factor affected not to lower the risk of PD.
=> [Discussion] In this study, we did not identify an association between gout and PD after adjusting for age, sex, and comorbidities. PD is a complex neurodegenerative disease and it can be affected by many other factors other than gout. Therefore, we investigated other potential factors affecting PD. A cox proportional hazard model revealed that old age, female sex, stroke, and hypertension were independent risk factors for PD.
Han et al [6] reported that the prevalence and incidence of PD were higher in women than in men. Other studies on Asian populations in Japan, China, and Taiwan revealed consistent results, with a reported female preponderance or no sex-related differences in the occurrence of PD [23-25]. In our study, the result identified female sex as a risk factor for PD, although only 7% of the study cohort was female. Conversely, studies in Europe and North/South America have reported opposing results, suggesting that sex differences may exist according to race/ethnicity [26-29].
We conducted sub-analysis by sex group for risk factors affecting PD using cox-proportional hazard model. The analysis result revealed that stroke was a still important risk factor and hypertension also tended to increase risk for PD in female group. Therefore, we suggest that controlling hypertension, a modifiable risk factor, and preventing stroke can be helpful to lower the risk of PD, especially in the females, who are the risk group.
=> [Discussion] In this study, the HR of dyslipidemia for PD risk was 0.6 [95% CI 0.52-0.68, p<0.001] in the whole patients, HR 0.6 in male group [95% CI 0.52-0.69, p<.0001] and HR 0.59 in the female group [95% CI 0.42-0.82, p=0.0017]. These results were consistent with recent two large cohort studies [32,33]. A possible explanation is as follow.